# New Generation Express View: An Artificial Intelligence Software Effectively Reduces Capsule Endoscopy Reading Times

**DOI:** 10.3390/diagnostics12081783

**Published:** 2022-07-22

**Authors:** Stefania Piccirelli, Alessandro Mussetto, Angelo Bellumat, Renato Cannizzaro, Marco Pennazio, Alessandro Pezzoli, Alessandra Bizzotto, Nadia Fusetti, Flavio Valiante, Cesare Hassan, Silvia Pecere, Anastasios Koulaouzidis, Cristiano Spada

**Affiliations:** 1Fondazione Poliambulanza Istituto Ospedaliero, 25124 Brescia, Italy; stefania.piccirelli@gmail.com (S.P.); alessandrabizzotto@gmail.com (A.B.); cristianospada@gmail.com (C.S.); 2Fondazione Policlinico Universitario A. Gemelli IRCCS, Università Cattolica del Sacro Cuore, 00168 Rome, Italy; 3Ospedale Santa Maria delle Croci, 48121 Ravenna, Italy; alessandromussetto@gmail.com; 4Ospedale di Feltre, 32032 Feltre, Italy; abellum@alice.it (A.B.); flavio.valiante@gmail.com (F.V.); 5Centro di Riferimento Oncologico, IRCCS, 33081 Aviano, Italy; rcannizzaro@cro.it; 6Division of Gastroenterology, University City of Health and Science University Hospital, 10121 Turin, Italy; pennazio.marco@gmail.com; 7Endoscopy Unit, Department of Gastroenterology, Sant’Anna University Hospital, 44121 Ferrara, Italy; a.pezzoli@ospfe.it (A.P.); n.fusetti@ospfe.it (N.F.); 8Endoscopy Unit, Department of Gastroenterology, IRCCS Humanitas Research Hospital, 20089 Rozzano, Italy; cesareh@hotmail.com; 9Digestive Endoscopy Unit, Fondazione Policlinico Universitario A. Gemelli IRCCS, 00168 Rome, Italy; 10Department of Medicine, Odense University Hospital Svendborg Sygehus, 5700 Svendborg, Denmark; akoulaouzidis@hotmail.com; 11Department of Clinical Research, University of Southern Denmark (SDU), 5230 Odense, Denmark; 12Surgical Research Unit, Odense University Hospital, 5000 Odense, Denmark; 13Department of Social Medicine and Public Health, Pomeranian Medical University, 70-204 Szczecin, Poland

**Keywords:** capsule endoscopy, automatic reading software, reading time, Express View

## Abstract

BACKGROUND: Reading capsule endoscopy (CE) is time-consuming. The Express View (EV) (IntroMedic, Seoul, Korea) software was designed to shorten CE video reading. Our primary aim was to evaluate the diagnostic accuracy of EV in detecting significant small-bowel (SB) lesions. We also compared the reading times with EV mode and standard reading (SR). METHODS: 126 patients with suspected SB bleeding and/or suspected neoplasia were prospectively enrolled and underwent SB CE (MiroCam^®^1200, IntroMedic, Seoul, Korea). CE evaluation was performed in standard and EV mode. In case of discrepancies between SR and EV readings, a consensus was reached after reviewing the video segments and the findings were re-classified. RESULTS: The completion rate of SB CE in our cohort was 86.5% and no retention occurred. The per-patient analysis of sensitivity, specificity, positive predictive value, negative predictive value, and diagnostic accuracy of EV compared to SR were 86%, 86%, 90%, 81%, and 86%, respectively, before consensus. After consensus, they increased to 97%, 100%, 100%, 96%, and 98%, respectively. The median reading time with SR and EV was 71 min (range 26–340) and 13 min (range 3–85), respectively (*p* < 0.001). CONCLUSIONS: The new-generation EV shows high diagnostic accuracy and significantly reduces CE reading times.

## 1. Introduction

Capsule endoscopy (CE) is a well-established method to explore the small bowel (SB). National and International guidelines recommend performing CE in case of suspected SB bleeding (SSBB), suspected or known Crohn’s disease, suspected SB tumors, inherited polyposis syndromes, and celiac disease [1,2]. Although continuous technological advances during the last three years improved CE image quality and battery life, CE reading remains time-consuming, with mean reading times of 50–75 min (ranging from 30 to 120 min) and based on the reader’s experience [3,4,5,6,7,8,9]. Reading requires the clinicians to pay and maintain attention for a long time without distractions [10]. A recent study showed that the fatigue of experienced readers significantly affects their accuracy, causing a decrease of 20% in lesion detection after the first CE study [11]. To shorten reading times, CE readers often increase the frame rate. However, this strategy inevitably leads to an increased and unacceptable miss rate by reducing significant lesion detection [12]. Innovative algorithms have been developed to shorten video evaluation by removing redundant images based upon their similarity, thus reducing reading time, such as RAPID^®^, QuickView (PillCam, Medtronic, Dublin, Ireland) [13,14,15,16,17,18], Omni Mode (Endocapsule, Olympus, Tokyo, Japan) [5,6], MiroViewTM, and Express View (MiroCam^®^, IntroMedic, Seoul, Korea) [8,19]. Dedicated gastroenterologists have accepted these technologies, as shown in a recent international survey by the I CARE (CApsule endoscopy Research) group. However, they also pointed out the need for CE readers to implement artificial intelligence (AI) systems further [20].

Express View (EV) is a reading software designed to reduce CE video by simultaneously applying a function that recognizes abnormal findings (suspected disease detection) and another function that enables the cut of insignificant redundant images (video summary). This software was initially validated by Saurin et al. [19] and by Gomes et al. [8] on 83 and 89 patients, respectively, showing 82.2–83.1% sensitivity for detection of clinically significant lesions. Recently, a third-generation artificial intelligence-based EV algorithm, working with a binary map and a computational 8-ray connectivity, was released. The abnormal detection function, based on a conventional machine learning (ML) algorithm that uses a feature descriptor set of color, texture, and shape, was reinforced and tunned with other lesions (bleeding, ulcer, vascular, polyp) to implement local features for ROI (region of interest). Moreover, the video summary function, which can convert the red–green–blue (RGB) color model into the hue–saturation–value (HSV) to further compare images and skip similar ones, was improved. Implementing the histogram comparison of the local binary pattern (LBP) format, the video summary allows the removal of meaningless images, providing a compression rate of the initial video of up to 95% (compared to 80% of EV version 2.0). This multicenter study aims to prospectively evaluate the diagnostic accuracy of the new-generation EV in a per-patient and per-lesion analysis and to evaluate its impact on reading time by comparing the time of reading in EV mode and standard mode.

## 2. Materials and Methods

### 2.1. Population and Procedure

Patients undergoing CE in 6 Italian centers (Brescia, Ravenna, Feltre, Aviano, Torino, Ferrara) were enrolled for this study between January 2019 and July 2020. Each center consecutively enrolled 20 patients, except for the coordinating center (Brescia), which consecutively enrolled 26 patients. All patients signed the required informed consent form. Indications for CE were SSBB (overt or occult) with negative bi-directional endoscopy and/or suspected SB neoplasia (based on symptoms and previous radiological tests) or polyposis. CE was performed with the MiroCam^®^1200 (IntroMedic, Seoul, Korea), which provides a 320 × 320 resolution, a field of view of 170°, and an image acquisition rate of 3 frames per second (fps). Each participating center was allowed to follow its pre-procedural preparation protocol if based on the current international guidelines [1,21]. Therefore, all patients had 2 L polyethylene–glycol (PEG)-based bowel prep as a split regimen, simethicone 30 min before capsule ingestion, and prokinetics administered orally in case the capsule was found in the stomach, 1 h after its ingestion, with real-time viewer check. 

CE was considered complete if the capsule reached the caecum within the recording time. All patients with a complete SB transit and those in whom the capsule entered the SB (even if not reaching the caecum) were included in the analysis. In addition, capsule retention data, defined as a capsule that was not excreted within 15 days, and complication rates, were collected.

### 2.2. Capsule Reading

Each video was initially evaluated in standard view by one experienced (>200 cases) local reader from the enrolling center (standard reading, SR). A single frame reading at 20–25 fps speed was used for stomach review, whereas the SB reading was performed at a maximum speed of 10–12 fps, as suggested by the ESGE Guidelines [21,22]. Once generated, anonymized videos were sent to the “coupled center”, chosen among the other centers, for the following blinded reading in EV mode (EV Version 3 2018). The shortened continuous video (removed by redundant frames) processed by the EV software was reviewed by the blinded reader with the same modalities of SR. For both standard and EV reading, the readers collected clinical information and findings in a case report form (CRF); the bowel cleansing was also reported using the QI (Quantitative Index) Brotz scale [23]. Positive findings were classified in terms of type, location, and relevance as P0-P2 (according to the Saurin classification) [24]. Reading time was defined as the time (in min) spent watching the video (in both modalities), including the time for selection of images, without considering the time spent to write the final report. Reading time for the SB examination was recorded and reported separately from the esophagus and gastric reading time.

### 2.3. Interpretation of the Results

A consensus meeting of designated experts among the principal investigators of the involved centers (AM, CS, and SP, each >400 CE reading experience) examined all cases by comparing the CRFs filled by the standard reader with those filled by the blinded reader in EV mode. In the case of multiple findings corresponding to different diagnoses and clinical relevance, the most significant lesion supporting the reported diagnosis was specified. Concordant cases for positive or negative findings (final diagnosis and/or detected lesions) were not re-evaluated. Conversely, in case of discrepancies between the two reading modes, a panel of experts re-evaluated the videos and adjudicated. Specifically, a face-to-face meeting was organized, and the 3 consensus panel members simultaneously re-evaluated the full videos of the discordant cases using both the standard and EV reading, and eventually reclassified the findings. The primary endpoint was to evaluate EV sensitivity in detecting SB lesions (P1 and P2, according to the Saurin classification) before and after consensus review with per-patient and per-lesion analysis. SR was first considered the reference standard and used as a comparison to measure the reading agreement. After consensus, EV accuracy was re-evaluated considering SR after consensus review the new reference standard. Secondary endpoints were (1) median reading time with EV and SR, (2) evaluation of EV accuracy in detecting significant SB findings (P1 and P2 lesions) before and after consensus, and (3) diagnostic yield (DY) of EV and SR when compared to consensus review (Figure 1).

### 2.4. Statistical Analysis

Reading time for EV and SR was expressed as a median (min and max value range). Means and standard deviations were used to describe continuous variables. Proportions were used to describe categorical variables. A Mann–Whitney test was used to compare the median reading time in EV mode and SR. EV accuracy was measured in terms of sensitivity (sens), specificity (spec), positive predictive value (PPV), negative predictive value (NPV), and diagnostic accuracy, all presented as percentages (95% confidence interval, CI) at the per-patient and per-lesion analysis. The final diagnosis was considered for the per-patient analysis. In the case of a CE video showing numerous (>10) repetitive SB lesions (including Crohn’s disease with numerous ulcerations, or diffuse angioectasias), these findings were considered as corresponding to one single lesion for the per-lesion analysis. Although variability exists in the literature, meta-analyses and larger studies suggest about a 60% DY for SB CE in patients with SSBB [2]. Therefore, by assuming a 60% prevalence of clinically relevant lesions, a sample size of 121 patients would estimate sensitivity of 90% of the new method with a 95% CI between 83% and 97%. Considering a failure rate of 5%, the total sample size was 126 patients.

## 3. Results

In the present study 126 patients were enrolled (mean age 67.6 ± 14.6; 64 women). Indications for CE were occult SSBB in 99 patients (78.6%), overt SSBB in 25 patients (19.8%), and suspected SB neoplasia in 2 patients (1.6%). Technical issues (i.e., recording was temporarily interrupted during capsule transit in the stomach) were reported in one; CE was complete in 109 out of 126 patients (86.5%). In 17 patients the capsule did not reach the caecum. Incomplete examination occurred in 11 out of 17 patients due to delayed SB transit and in 2 patients due to SB strictures (n = 1 duodenal fibrotic ulcer; n = 1 jejunal adenocarcinoma). In the remaining four cases, the incomplete examination was related to delayed gastric transit (last CE image was acquired in the stomach). No case of capsule retention occurred. In one case, the capsule was endoscopically retrieved after 4 days. No adverse events were reported. Mean gastric and SB transit times were 40.4 min (range 0–264) and 293.3 min (range 23–690), respectively (Table 1). A total of 98 out of 122 patients with SB evaluation had a QI Brotz scale ≥ 6. Mean QI Brotz scores given by readers in standard mode and EV mode were 7.11 ± 1.93 and 6.96 ± 1.86, respectively.

### 3.1. Per-Patient Analysis

One hundred twenty-two (96.8%) CEs were included in the analysis. Before consensus, SR and EV agreed with the final diagnosis in 105 cases (n = 62 positive for SB finding and n = 43 negative). The remaining 17 cases had discordant results. In particular, 10 had a positive SR (SR+) and negative EV reading (EV−); 7 had a negative SR (SR-) and positive EV reading (EV+). Before consensus, EV sensitivity, specificity, PPV, and NPV were 86%, 86%, 90%, and 81%, respectively. The overall EV diagnostic accuracy was 86% (Table 2). After consensus, the 17 discordant cases were reclassified as following: All seven cases SR-EV+ were reclassified as positive at both SR and EV mode (SR+EV+); of the remaining 10 SR+EV− cases, seven were reclassified SR+EV+ and one was reclassified as SR−EV−; and the remaining two cases were confirmed to have discordant reading (SR+EV−) since the EV software inappropriately cut the findings responsible for the main diagnosis (an ileal hemangioma and an ileal erosion). As a result, 78 patients were reclassified as having a final diagnosis affecting the SB, whereas 44 patients had no significant pathological SB findings (Table 3 and Figure 2). After consensus, EV sensitivity, specificity, PPV, and NPV were 97%, 100%, 100%, and 96%, respectively, with an overall diagnostic accuracy of 98% (Table 2). The accuracy of EV and SR after consensus (representing the reference standard) are reported in Table 4.

### 3.2. Per-Lesion Analysis

At the per-lesion analysis performed on 122 patients, 119 out of 164 lesions were detected by both SR and EV, whereas the remaining 45 lesions were reported discordantly. In particular, 29 lesions resulted in SR+ and EV–, and 16 lesions were SR– and EV+. Before consensus, EV sensitivity, specificity, PPV, NPV, and diagnostic accuracy were 80%, 78%, 88%, 66%, and 80%, respectively. After the consensus review, the 45 discordant findings were reclassified as follows: All 16 SR–EV+ lesions were reclassified as positive at both readings, whereas of the remaining 29 SR+EV– lesions, 23 were reclassified as SR+EV+, 3 lesions were reclassified as SR–EV–, and in 3 cases the EV reading was unable to diagnose the lesion (SR+EV–). The lesions missed were an ileal angiodysplasia, an ileal hemangioma, and an ileal erosion. To note, in two out of three cases of missed lesions, the quality of preparation was judged inadequate. As a result, 158 out of 161 lesions were reclassified as visible at both SR and EV reading, whereas three lesions were non-visible at EV reading (Table 3). After consensus, EV sensitivity, specificity, PPV, and NPV were 98%, 100%, 100%, and 95%, respectively, with an overall diagnostic accuracy of 99% (Table 2). The accuracy of EV and SR after consensus (representing the reference standard) is reported in Table 4. 

### 3.3. Reading Time

The median reading time in standard mode and EV mode was 71 min (range 26–340 min) and 13 min (range 3–85 min), respectively (*p* < 0.0001). Applying the EV algorithm, images were reduced by 94.8%, with 11,737,264 in standard mode and 609,649 after the EV cut. Concerning small-bowel cleansing, the mean and median values of the Brotz score given by the first and second readers were comparable (mean 7.1 ± 1.9, median 7, and mean 6.9 ± 1.9, median 7, respectively) and resulted in globally inadequate cleansing.

## 4. Discussion

In this multicenter, prospective trial, the EV software was demonstrated to be highly sensitive in the detection of significant (P1 + P2 according to the Saurin classification) [24] SB lesions and consistently shortening reading time. Recently, Saurin and Gomes evaluated a previous version of the EV software and reported a per-patient sensitivity of EV reading ranging between 82.2% [19] and 83.1% [8], with a software sensitivity of 94.2% [19]. The new-generation EV software tested in the present study confirmed an even higher per-patient sensitivity, both in terms of reading in EV mode (88% sensitivity) and software sensitivity, equal to 98%. 

Interestingly, EV software showed lower accuracy for ileal lesions. This result is in line with previous evidence by Gomes et al. [8], who showed a reduced detection of ileal lesions when compared to non-SB lesions and those located in the duodenum and the jejunum. The reason for such a result is still unclear. However, considering the lesions missed by the software, we noticed that the patient with a missed ileal erosion (P2) had another ileal erosion (P1) that was detected by the software, and the patient with an ileal angiodysplasia (P1) had other two ileal angiodysplasias (P2 and P1) that were detected by the EV software. This might suggest that in case of multiple lesions, the software may detect at least one among various similar abnormal findings, thus not negatively impacting the final diagnosis and patient management. 

In addition, an inadequate cleansing level might contribute to incorrect lesion detection by the EV software. In the present series, the small number of missed lesions precludes the evaluation of the potential role of the quality of preparation on the software accuracy. Nevertheless, it should be noted that in two out of three cases of missed lesions the quality of preparation was considered inadequate, suggesting that the quality of preparation might play a role. Regarding secondary outcomes, EV proved to shorten the reading time significantly. Thirteen minutes of median reading time represents a relevant improvement that further reduces the reading time that was observed in previous studies evaluating CE reading using EV [8,19].

Notably, in the present study a greater percentage of time-saving compared to that obtained by Saurin (51.5%) [19] and Beg (57.9%) [5] was observed (82.1%). This difference could be explained by a longer standard reading time that might lead to overestimating the reading time saved by using the EV mode. However, CE reading within a research protocol may be more accurate and careful, and consequently more time-consuming, irrespective of the reading mode. Hence, it is uncertain whether the same gain could be achieved in routine clinical practice. Other limitations should be underlined. First, a consensus review was performed by re-evaluating only video segments that included the missed lesion (either at SR or EV reading). 

Although we assumed that the EV software did not miss any lesion if readers in standard mode and EV mode agreed with the final diagnosis, such an assumption was never tested and needs to be confirmed in further trials. Second, inter-observer agreement among readers in different modes was not evaluated. AI-based software that shortens the reading time is likely to play a role in reducing the impact of fatigue on the miss rates and, ultimately, in increasing the inter-observer agreement. Studies need to be specifically designed to prove this hypothesis. Third, our study population does not represent all indications for CE since only patients with SSBB and/or suspected SB neoplasia were included. For instance, even if SSBB remains the main indication of SBCE, suspected or known Crohn’s disease represents a rising indication for CE, and EV accuracy in the detection of erosions and ulcers should be evaluated in further studies [25,26,27]. To note, the capsule does not allow pathological confirmation of findings. In our study, no further endoscopic characterization of findings was performed, and this is a limitation that should be considered for future trials. Finally, the impact of bowel cleansing on software accuracy was assessed poorly in our study. 

This evaluation might be crucial in determining the software accuracy in the field of automatic reading algorithms. Nevertheless, the qualitative and quantitative scores currently available to define SB cleansing are unreliable or lack reproducibility [28]. AI-based software to automatically assess bowel cleansing may overcome this issue in the near future [29,30,31,32]. In conclusion, CE reading is still highly time-consuming. It requires the reader to have prolonged and high concentration in order not to affect CE DY. The results of the present study suggest that the AI system Express View, based on 11 conventional machine learning algorithms and designed by IntroMedic, is highly sensitive in detecting small-bowel lesions and significantly reduces CE reading without affecting the DY.

## Figures and Tables

**Figure 1 diagnostics-12-01783-f001:**
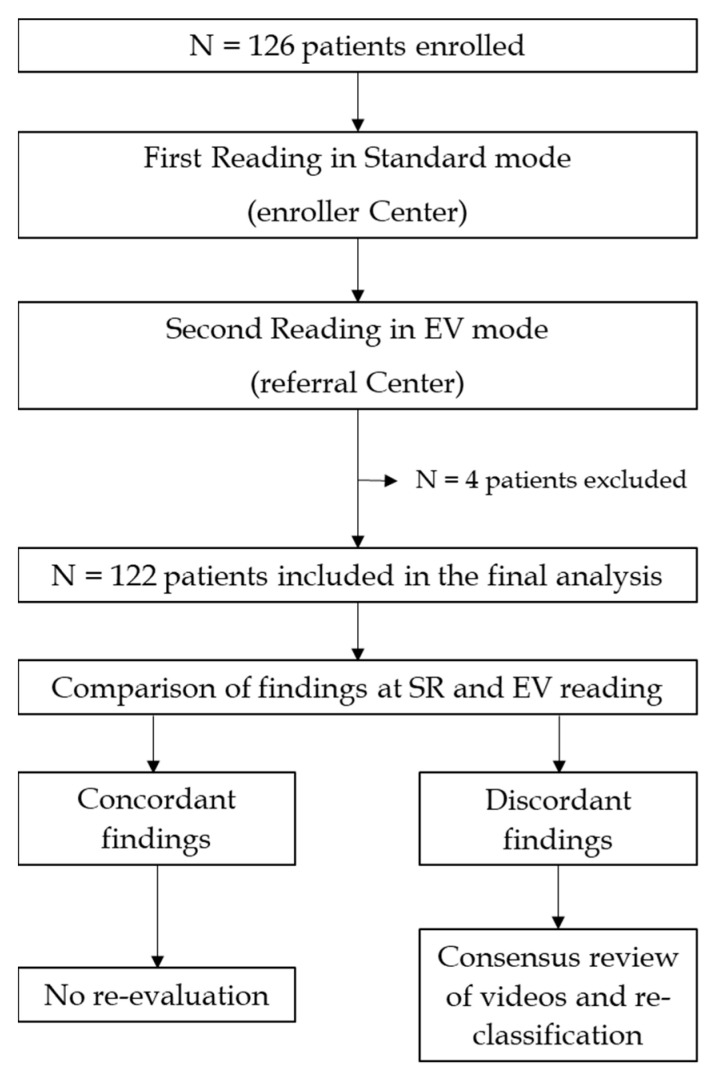
Study design.

**Figure 2 diagnostics-12-01783-f002:**
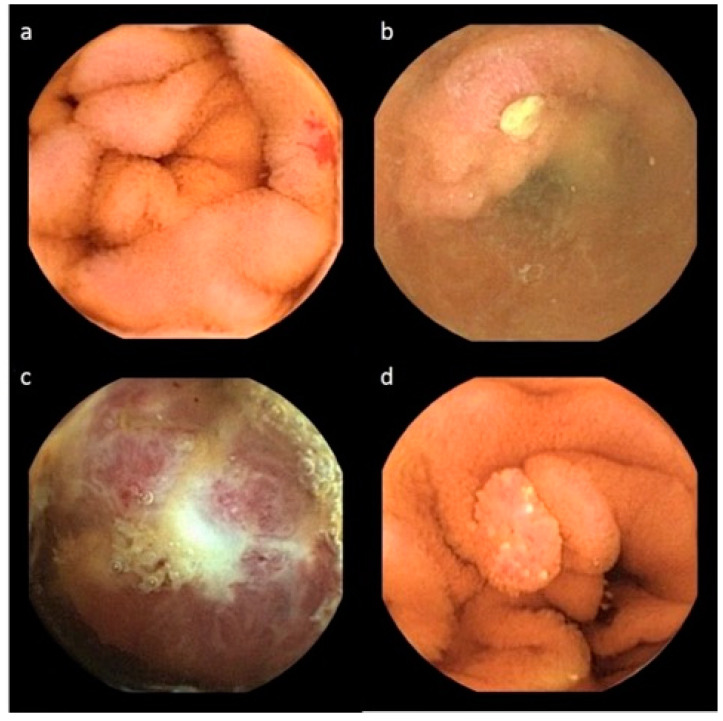
Capsule endoscopy findings. (**a**) Jejunal angiodysplasia; (**b**) ileal ulcer; (**c**) jejunal tumor; (**d**) jejunal polyp.

**Table 1 diagnostics-12-01783-t001:** Study population and characteristics.

Parameters	Value
Female, n (%)	64 (50.8)
Age (mean ± SD years)	67.6 ± 14.6
**Indications**	**N. of patients**
IDA	83
FOBT+	16
Overt bleeding	25
Suspected neoplasia	2
Total	126
**Capsule Examination**	**N. of patients**
Completion (reach of the caecum)	109
Incomplete examination	17
Delayed SBTT	11
Small bowel stricture	2
Delayed GTT	4
Retention	0

**Table 2 diagnostics-12-01783-t002:** Accuracy of Express View (EV) mode compared with standard reading (SR) before and after consensus review.

	Per-Patient Analysis		Per-Lesion Analysis	
	EV vs. SR (Before Consensus)	EV vs. SR (After Consensus)	EV vs. SR (Before Consensus)	EV vs. SR (After Consensus)
**Sensitivity**	86% (75.9–93.1)	97% (91–99.7)	80% (73.1–86.5)	98% (94.7–99.6)
**Specificity**	86% (73.3–94.2)	100% (92–100)	78% (66.4–86.7)	100% (93.9–100)
**PPV**	90% (80.2–95.8)	100% (95.3–100)	88% (81.5–93.1)	100% (97.7–100)
**NPV**	81% (68–90.6)	96% (85.2–99.5)	66% (54.8–75.8)	95% (93.9–100)
**Diagnostic accuracy**	86% (78.6–91.6)	98% (94.2–99.8)	80% (73.6–84.7)	99% (96.1–99.7)

IC 95%. EV = Express View; SR = standard view; PPV = positive predictive value; NPV = negative predictive value.

**Table 3 diagnostics-12-01783-t003:** Capsule endoscopy findings.

			Per-Patient Analysis (Final Diagnosis)	Per-Lesion Analysis (Lesions)
**Small Bowel**	**Angiodysplasia**			
	Unbleeding	Duodenum	n = 3	n = 8
		Jejunum	n = 36	n = 74
		Ileum	n = 6	n = 18
	Bleeding	Jejunum	n = 1	n = 1
	**Erosions**	Duodenum	n = 3	n = 5
		Jejunum	n = 4	n = 10
		Ileum	n = 5 (n = 2 isolated and n = 3 multiple aftoid erosions)	n = 11
	**Polyps**	Duodenum	n = 1	n = 1
		Jejunum	n = 3	n = 4
	**Red spots**	Duodenum		n = 1
		Jejunum	n = 2	n = 7
		Ileum	n = 2	n = 3
	**Blood in lumen**	Duodenum	n = 1	n = 1
		Jejunum	n = 2	n = 2
	**Ulcerated lesion**			
	Stricturing/substricturing	Duodenum	n = 1	n = 1
		Jejunum	n = 2	n = 2
		Ileum	n = 1	n = 1
	Non-stricturing	Jejunum	n = 1	n = 1
	**Ulcers**	Duodenum		n = 1
		Jejunum		n = 1
		Ileum	n = 2	n = 5
	**Suspected submucosal mass**	Jejunum		n = 1
		Ileum	n = 1	n = 1
	**Hemangioma**	Ileum	n = 1	n = 1
**Extra SB**	**Angiodysplasia**	Stomach	n = 1	n = 3
		Colon	n = 5	n = 10
	**Ulcer**	Stomach	n = 1	n = 1
	**Erosions**	Stomach		n = 1
	**GAVE**	Stomach		n = 1
	**Total**		**n = 78**	**n = 161**

**Table 4 diagnostics-12-01783-t004:** Diagnostic accuracy of readers in EV mode and SR when both are compared to consensus review.

	Per-Patient Analysis		Per-Lesion Analysis	
	EV vs. Consensus	SR vs. Consensus	EV vs. Consensus	SR vs. Consensus
**Sensitivity**	88% (79.2–94.6)	91% (82.4–96.3)	85% (79–90.5)	90% (84.4–94.2)
**Specificity**	100% (90.5–100)	98% (88–100)	100% (94.3–100)	95% (85.9–98.3)
**PPV**	100% (94.8–100)	99% (92.5–100)	100% (97.3–100)	98% (94.2–99.6)
**NPV**	83% (66.1–90.6)	86% (73.3–94.2)	73% (62.2–82)	78% (66.4–86.7)
**Diagnostic accuracy**	93% (86.5–96.6)	93% (87.5–97.2)	90% (84.7–93.3)	91% (86.8–94.7)

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
