# Peer review of "New Generation Express View: An Artificial Intelligence Software Effectively Reduces Capsule Endoscopy Reading Times"

_diagnostics, 2022, doi:10.3390/diagnostics12081783_

Round 1

Reviewer 1 Report

The study tested diagnostic accuracy of new generation EV of small bowel capsule. It reflects recent advancement in the field of diagnostic AI.

I only have few questions.

1. in page 2 under materials and methods section, it is stated as " Indications for CE were (overt or occult) SSBB ..." wouldn't it be better to place (over or occult) after SSBB rather than before? 

2. Among 7 cases that were initially SB +/EV - which turned out to be EV + after reviewing, it is still unclear to me how machine interpretation can have different outcome. The same goes for all disconcardant cases. Sensitivity using pre-consensus results does not seem so different from previous studies using older model. Does that mean the EV needs adjustment before being applied to clinical settings?

Author Response

The study tested diagnostic accuracy of new generation EV of small bowel capsule. It reflects recent advancement in the field of diagnostic AI.

 I only have few questions.

  1. in page 2 under materials and methods section, it is stated as " Indications for CE were (overt or occult) SSBB ..." wouldn't it be better to place (over or occult) after SSBB rather than before? 

We thank the reviewer to have pointed it out. We modified it in the manuscript accordingly.

  1. Among 7 cases that were initially SB +/EV - which turned out to be EV + after reviewing, it is still unclear to me how machine interpretation can have different outcome. The same goes for all disconcardant cases. Sensitivity using pre-consensus results does not seem so different from previous studies using older model. Does that mean the EV needs adjustment before being applied to clinical settings?

Thank the reviewer for the comment. In the text, we discuss both the sensitivity of the informatics algorithm (EV), and the sensitivity of the readings performed using (or not) EV mode. In the reclassification of discordant cases, it is not the machine interpretation to have different outcome, but the readers that may have missed or not a finding. Most of SR+/EV- cases were false negatives (the readers missed the finding). Only two cases SR+/EV- were true negatives (the software missed the finding).

Regarding the sensitivity of EV, previous studies reported a good sensitivity of EV. Our study demonstrated a further improvement of the software accuracy (98% vs 94.2% of Saurin) and a further improvement in terms of accuracy of readers using EV mode (88% vs 82.2%). Moreover, in our study, we found that EV and SR had comparable accuracies (88% and 91%, respectively) with a significant reduction of reading time (13 minutes instead of 71, mean value). If we look at the results of Saurin, we see that the difference in terms of accuracy between EV and SR was higher (82.2% and 93.3%), and EV was less accurate than the new version we evaluated (- 6%).

Reviewer 2 Report

This is an interesting study on the efficacy of artificial intelligence (Express View: EV) for capsule endoscopy. The efficacy of EV has been assessed in comparison to diagnosis by standard reading, and consensus diagnosis. The authors have demonstrated that the diagnostic yield with EV is nearly equivalent to the diagnostic yield with standard reading, with significantly shorter reading times. I have a few comments on the study design and interpretation of results.

1. The authors mention that this is a prospective study. How was the sample size determined? Was there any logical reasoning for the sample size? This should be clarified in the methods.

2. Also related to comment 1, The authors state that "consecutive patients undergoing CE" were enrolled (2.1. Population and procedure). It seems that the number of patients in each center seems to be too similar for a multicenter prospective study enrolling all consecutive patients. If the study was designed so that each center enrolled 20 consecutive patients respectively, and not consecutive patients as a multicenter prospective study, the wording should be corrected in the methods.

3. There were 2 golden standards for accuracy of capsule endoscopy: diagnosis with standard reading, and consensus diagnosis by a panel of experts. Both of these golden standards are so-called "endoscopic" diagnoses with no pathological confirmation. Were these "endoscopic" diagnoses reconfirmed by any other modalities, such as biopsies with double-balloon enteroscopy? The reason for this question is because the authors mention that "the software may detect at least one among various similar pathological findings" in the 2nd paragraph of the discussion. If there is no pathological confirmation of any of the diagnoses, any mention of "pathological findings" seems to be a leap of logic, and should be removed. In addition, if there was no confirmation of pathology, this should be listed as a limitation of this study.

4. Also related to comment 3, the authors state that diagnosis with EV is "accurate". However, this is based on the assumption that "differential" diagnosis with capsule endoscopy is accurate. The authors should include proper references that support this assumption. Otherwise, it seems that the authors have only demonstrated the high "diagnostic yield" of EV, not the "accuracy". The authors may need to reconsider which word better expresses the results of this study.

Author Response

This is an interesting study on the efficacy of artificial intelligence (Express View: EV) for capsule endoscopy. The efficacy of EV has been assessed in comparison to diagnosis by standard reading, and consensus diagnosis. The authors have demonstrated that the diagnostic yield with EV is nearly equivalent to the diagnostic yield with standard reading, with significantly shorter reading times. I have a few comments on the study design and interpretation of results.

  1. The authors mention that this is a prospective study. How was the sample size determined? Was there any logical reasoning for the sample size? This should be clarified in the methods.

Thank you for the comment. Please, find the sample size analysis described in the paragraph 2.4. Although variability exists in literature, meta-analyses and larger studies suggest about a 60% DY for SB CE in patients with SSBB. [2] Therefore, by assuming a 60% prevalence of clinically relevant lesions, a sample size of 121 patients would estimate sensitivity of 90% of the new method with a 95% CI between 83% and 97%. Considering a failure rate of 5%, the total sample size was 126 patients.

  1. Also related to comment 1, The authors state that "consecutive patients undergoing CE" were enrolled (2.1. Population and procedure). It seems that the number of patients in each center seems to be too similar for a multicenter prospective study enrolling all consecutive patients. If the study was designed so that each center enrolled 20 consecutive patients respectively, and not consecutive patients as a multicenter prospective study, the wording should be corrected in the methods.

We are grateful to the reviewer for having noticed this error. We modified it accordingly in the manuscript.

  1. There were 2 golden standards for accuracy of capsule endoscopy: diagnosis with standard reading, and consensus diagnosis by a panel of experts. Both of these golden standards are so-called "endoscopic" diagnoses with no pathological confirmation. Were these "endoscopic" diagnoses reconfirmed by any other modalities, such as biopsies with double-balloon enteroscopy? The reason for this question is because the authors mention that "the software may detect at least one among various similar pathological findings" in the 2nd paragraph of the discussion. If there is no pathological confirmation of any of the diagnoses, any mention of "pathological findings" seems to be a leap of logic, and should be removed. In addition, if there was no confirmation of pathology, this should be listed as a limitation of this study.

Thank you for the commment, this is an important point. The software has the purpose to cut similar (unsignificant) findings to shorten the video, without removing significant abnormal findings. In this case, the hypothesis is that it may have cut similar abnormal findings.  Then, I would rather replace “pathological” with “abnormal” than leaving just “similar findings.

As you suggested, we added the limitation of defining pathology in the text.

  1. Also related to comment 3, the authors state that diagnosis with EV is "accurate". However, this is based on the assumption that "differential" diagnosis with capsule endoscopy is accurate. The authors should include proper references that support this assumption. Otherwise, it seems that the authors have only demonstrated the high "diagnostic yield" of EV, not the "accuracy". The authors may need to reconsider which word better expresses the results of this study.

Thank you for the precious suggestion. We agree that the term “accurate” may be inappropriate and replaced it with the term “sensitive”. We used this in all sentences regarding accuracy in the manuscript. 

Round 2

Reviewer 2 Report

The authors have sufficiently addressed all comments.